

# Preference classes in society for coastal marine protected areas

Ana Ruiz-Frau[1,2], James M. Gibbons[3], Hilmar Hinz[1],
Gareth Edwards-Jones[3,†] and Michel J. Kaiser[4]

[1] Department of Marine Ecology, Instituto Mediterráneo de Estudios Avanzados, Esporles, Spain
[2] School of Ocean Sciences, College of Environmental Sciences and Engineering, Bangor University, Bangor, UK
[3] School of Natural Sciences, College of Environmental Sciences and Engineering, Bangor University, Bangor, UK
[4] The Lyell Centre, School of Energy, Geosciences, Infrastructure and Society, Heriot-Watt University, Edinburgh, UK
† Deceased author.

Corresponding author
Ana Ruiz-Frau,
anaruiz@imedea.uib-csic.es

## ABSTRACT

Marine protected areas (MPAs) are increasingly being used as conservation tools in the marine environment. Success of MPAs depends upon sound scientific design and societal support. Studies that have assessed societal preferences for temperate MPAs have generally done it without considering the existence of discrete groups of opinion within society and have largely considered offshore and deep-sea areas. This study quantifies societal preferences and economic support for coastal MPAs in Wales (UK) and assesses the presence of distinct groups of preference for MPA management, through a latent class choice experiment approach. Results show a general support for the protection of the marine environment in the form of MPAs and that society is willing to bear the costs derived from conservation. Despite a general opposition toward MPAs where human activities are completely excluded, there is some indication that three classes of preferences within society can be established regarding the management of potentially sea-floor damaging activities. This type of approach allows for the distinction between those respondents with positive preferences for particular types of management from those who experience disutility. We conclude that insights from these types of analyses can be used by policy-makers to identify those MPA designs and management combinations most likely to be supported by particular sectors of society.

## INTRODUCTION

The marine environment provides society with a wide range of goods and services that are essential for the maintenance of our economic and social wellbeing (*MEA, 2005*; *Liquete et al., 2013*; *Costanza et al., 2014*). The recognition of the effects of anthropogenic activities on marine ecosystems has led to increasing conservation initiatives globally. Marine protected areas (MPAs) are among the most important tools available for

achieving global marine conservation targets, which are recognized both at international and European levels (*OSPAR, 2003*; *CBD, 2008*; *MSFD, 2008*).

Although the role of MPAs in the recovery of fish stocks and fisheries management remains an issue of debate (*Kaiser, 2005*; *Stefansson & Rosenberg, 2006*; *Hilborn, 2018*), it is clear that the establishment of MPAs has positive benefits for habitat restoration and biodiversity conservation within the boundaries of the MPA (*Halpern, 2003*; *Blyth-Skyrme et al., 2006*). However, the creation and enforcement of MPAs is costly (*Balmford et al., 2004*) and despite their potential benefits, their designation is often complex both legally and socially. This is because the closure of portions of the sea to human activities has impacts on those sectors of society directly affected by the closures, and not all of these impacts are perceived as positive. However, if designed carefully, MPAs can achieve a balance between marine conservation and socio-economic objectives (*Klein et al., 2008*; *Ruiz-Frau et al., 2015a*, *2015b*). Consequently, the design of MPAs is better addressed from an interdisciplinary perspective that is able to provide insights into the range of potential consequences of implementation. If MPAs are to successfully achieve their conservation objectives, then the biological principles of good reserve design need to have a strong influence on the designation process (*Roberts et al., 2003*), unfortunately, this is not always the case (*Caveen et al., 2013*, *2015*). In addition, conservation objectives cannot be met without support from members of local communities, resource users, and policy makers (*Moore et al., 2004*). Through the acknowledgement of the role of the marine environment as supplier of ecosystem services and benefits fundamental for the maintenance of human wellbeing (*MEA, 2005*), there is increasing pressure to engage stakeholders and in general members of society into marine and coastal planning (*EU, 2001*; *Epstein, Nenadovic & Boustany, 2014*; *Christie et al., 2017*). This paper focuses on expanding current knowledge on how the general public perceives and values the conservation of the marine environment and how distinct opinion groups can be established in society based on MPA management preferences. To do this, a case study using discrete choice experiment (DCE) methodology to assess society's preferences for the establishment of MPAs around the coast of Wales (UK) is undertaken.

Discrete choice experiments are survey-based methodologies where respondents are asked to choose their most preferred alternative among a set of hypothetical alternatives. Each alternative is characterized by the same bundle of attributes, however, the alternatives differ in the levels displayed by the attributes. Through the analysis of responses, the marginal rate of substitution between any pair of attributes that differentiate the alternatives can be determined. If one of the attributes has a monetary price attached to it, it is then possible to compute the respondent's willingness to pay (WTP) for the other attributes (*Hanley, Wright & Adamowicz, 1998*; *Liu et al., 2010*).

The body of literature using DCEs to determine the economic preferences and value that society attaches to the conservation of the marine environment through MPAs is rapidly growing (*Torres & Hanley, 2017*). A high proportion of studies, however, have focused on tropical areas and on coral habitats, which are highly charismatic and

might attract higher WTP from the public (*Mwebaze & MacLeod, 2013*; *Rogers, 2013*; *Rolfe & Windle, 2012*; *Torres & Hanley, 2016*), however cultural aspects need to be taken into account when considering studies from different locations, as charismatic species are not always the main drivers in the WTP for biodiversity conservation (*Ressurreição et al., 2012*). Similarly, studies on temperate areas are also increasing albeit at a lower pace (*McVittie & Moran, 2010*; *Wattage et al., 2011*; *Börger et al., 2014*; *Jobstvogt, Watson & Kenter, 2014*; *Jobstvogt et al., 2014*; *Kermagoret et al., 2016*; *Börger & Hattam, 2017*). In general, these studies indicate that society values, and is willing to support, the additional economic costs associated with conservation. As an example, *Jobstvogt, Watson & Kenter (2014)* showed high WTP values (£70–£77) for the protection of deep sea biodiversity while *McVittie & Moran (2010)* found values of similar magnitude for halting the loss of marine biodiversity in UK waters (£21–£34). However, the focus of the DCE studies on temperate MPAs has largely been on offshore and deep-sea areas (*Wattage et al., 2011*; *Börger et al., 2014*; *Jobstvogt et al., 2014*; *Kermagoret et al., 2016*; *Börger & Hattam, 2017*), on the WTP for the protection of charismatic species such as marine mammals through the use of MPAs (*Boxall et al., 2012*; *Batel, Basta & Mackelworth, 2014*) or on particular segments of society such as divers (*Sorice, Oh & Ditton, 2007*; *Jobstvogt, Watson & Kenter, 2014*). Additionally, those studies that considered society as a whole did not explore the existence of discrete opinion groups with distinct preferences for the management of MPAs and how this might be linked to particular socio-demographic characteristics and attitudinal aspects, as suggested by *Börger & Hattam (2017)* in a study on offshore areas. We argue that this type of information can be highly relevant for policy-makers during an MPA design process in order to enhance societal support.

The present study focuses on Wales in the UK, a region with a long coastline (approximately, 2,700 Km) and strong historic connections to the sea, where Government developed a Marine and Coastal Access Act 2009 in which it commits to "establishing an ecologically coherent, representative and well-managed network of MPAs" taking into account "environmental, social, and economic criteria" (*DEFRA, 2009*). In Wales, comprehensive information is available for the distribution of biophysical and ecological factors, however, information on how much the public values the conservation of the marine environment or on the support for MPAs in the area is scarce.

This case study offers an assessment of societal support for coastal MPAs located in temperate areas and analyses the assumption that there is preference heterogeneity in society for the type of protection of the marine environment and that discrete classes of preferences can be established through a DCE. Additionally, the focus of the study is on coastal waters for which people might be more familiar with and might have a greater sense of attachment in comparison to offshore areas and therefore preferences might differ.

## METHODS
The economic value associated with changes in the size and uses allowed within the boundaries of a temperate coastal MPA network were estimated using a DCE.

DCE data were collected using questionnaires. Heterogeneity in societal preferences for MPAs was estimated with a latent class (LC) choice experiment model (*Train, 2009*).

## Choice experiments econometrics

The economic framework for DCE lies in Lancaster's theory of consumer choices (*Lancaster, 1966*), which assumes that the utility of a good can be decomposed into the utilities of the characteristics of that good and as a result consumers' decisions are determined by the utility of the attributes rather than by the good itself. The econometric basis for DCE is provided by the random utility theory framework, which describes consumers' choices as utility maximization behaviors. Through the analysis of DCE data, marginal values for the attributes of a good or individual's WTP can be calculated (*Hensher, Rose & Greene, 2007*). However, DCE approaches remain controversial because of their hypothetical nature and the contested reliability of their results (*Hausman, 2012*), although it has been concluded that DCE remains useful for non-market valuation, its results should be used with caution (*Rakotonarivo, Schaafsma & Hockley, 2016*).

Discrete choice experiments can be analyzed using different models. Due to its simplicity, the multinomial logit model (MNL) is the most widely used. This model has important limitations; specifically, it assumes independence of irrelevant alternatives and it assumes homogeneous preferences for all respondents (*Hausman & McFadden, 1984*). However, within society preferences are heterogeneous and the ability to account for this variation allows the estimation of unbiased models that provide a better representation of reality. Random parameter logit models (RPL) and LC logit models relax the limitations of standard logit by allowing random taste variation and unrestricted substitution patterns in their estimation. The RPL allows utility parameters to vary randomly across individuals while in the LC formulation preference heterogeneity is captured by simultaneously assigning individuals into latent segments or "classes" while estimating a choice model. Within each LC, preferences are assumed homogeneous, but these can vary between classes (*Boxall & Adamowicz, 2002*; *Scarpa & Thiene, 2005*; *Colombo, Hanley & Louviere, 2009*). RPL approaches might not reveal the existence of classes since they are constrained by the assumed distribution across individuals, potentially hiding discrete groups. Model fit criterion measures were calculated for all models to assess their suitability to see which approach was most supported.

The utility (U) of a good consists of a known or systematic component (V) and a random component ($\varepsilon$) which is not observable by the researcher. The systematic component of utility can be further decomposed into the specific attributes of the good ($\beta X$), which in this case is a policy for the establishment of MPAs. Thus, the utility that respondent $n$ derives from a certain MPA alternative $i$ is given by:

$$U_{in} = \beta_{in} + \varepsilon_{in} \tag{1}$$
The probability that an individual $n$ will choose MPA alternative $i$ from a set of $J$ alternatives is equal to the probability that the utility derived from $i$ is greater than the utility derived from any other alternative:

$$\text{Prob}_{\text{in}} = \text{Prob}(U_{\text{in}} > U_{\text{jn}}) \; \forall j \in J \tag{2}$$

Assuming the random term to be independent and identically distributed according to a type I extreme value distribution, the probability that respondent chooses alternative $i$ in choice occasion $q$ is a standard MNL (*McFadden, 1974*):

$$L_{\text{n}}(i, q|\beta_{\text{n}}) = \frac{\exp(\beta_{\text{n}} X_{inq})}{\sum_{j}^{J} \exp(\beta_{\text{n}} X_{jnq})} \tag{3}$$

If is the respondent's chosen alternative in choice occasion and is the sequence of choices in Q choice occasions then the joint probability of the respondent's choices is the product of the standard logits:

$$\text{prob}(z_n|\beta_{\text{n}}) = L(z_{n1}, |\beta_{\text{n}})...L(z_{nQ}, Q|\beta_{\text{n}}) \tag{4}$$

The term $\beta_n$ is not directly observable, only its density is assumed to be known, where represents the parameters of the distribution. In RPL and LC models the unconditional probability of the respondent's sequence of choices is the integral of Eq. (4) over all possible values of $\beta_n$ determined by the population density of $\beta_n$:

$$\text{Prob}(z_n|\theta) = \text{prob}(z_n|\beta_{\text{n}})f(\beta_{\text{n}}|\theta)d\beta_{\text{n}} \tag{5}$$

The distribution of $\beta$ will determine the type of model to be used. If $\beta$ is continually distributed it will result in a RPL (*McFadden & Train, 2000*) while if the coefficients are discretely distributed and class membership is homogeneous it results on a LCM, where $\beta$ takes values for each class.

The log-likelihoods for both specifications are determined by:

$$L(\theta) = \sum_{n}^{N} \ln \text{Prob}(z_{\text{n}}) \tag{6}$$

Since the choice probability in the RPL does not have a closed form the expression has to be approximated using simulation (*Train, 2009*). Repeated draws of $\beta$ are taken from its density. For each draw, the product of logits is calculated and the results are averaged across draws. In this study, Halton intelligent draws have been used for the simulation since they have been found to provide greater accuracy than independent random draws in the estimation of RPL models (*Train, 2009*).

$$L_{\text{RPL}}(\theta) = \sum_{n=1}^{N} \ln \left[ \frac{1}{D} \text{Prob}(z_n|\beta^{\text{d}}) \right] \tag{7}$$

where $D$ is the number of draws and $\beta^d$ is the $d$th draw. For a LCM with C LCs, the log-likelihood function is given by:

$$L_{\text{LCM}}(\theta) = \sum_{n}^{N} \ln \left[ \sum_{C=1}^{C} \text{prob}(c)\text{Prob}(z_n|\beta_{\text{c}}) \right] \tag{8}$$

where Prob($c$) has a MNL form and is the probability of respondent $n$ belonging to class $c$ and $\beta_c$ represents a vector of class specific coefficients.

Welfare estimates can be derived from the models, they are calculated in the form of WTP using the formula:

$$WTP = \frac{\beta_a}{\beta_c} \tag{9}$$

where $\beta_a$ is the coefficient of the attribute of interest and $\beta_c$ is the negative of the coefficient of the monetary variable.

## Area of study

The study focused around the coastal waters of Wales (UK) (Fig. 1), prior to the initiation of formal Government consultation in late 2009. Here, we define Welsh coastal waters as those within the 12 nm territorial limit. In 2009, 32% of Welsh territorial waters were protected under a range of European and UK designations (Marine Nature Reserve, Special Area of Conservation, Special Protection Area and Site of Special Scientific Interest). However, existing designations were limited in terms of the species, habitats, or areas that were afforded protection and also the level of protection these different designations offered. At the time of writing, none of the designated areas were fully protected from human activities.

In the UK, the Marine and Coastal Access Act 2009 (DEFRA, 2009) provided the legislative powers necessary for the implementation of marine conservation zones (MCZs). Back in 2009 in Wales, the MCZ designation was to be primarily used to establish highly protected marine reserves (HPMRs), these are sites that are generally protected from extraction and deposition of living and non-living resources, and all other damaging or disturbing activities. The aim to establish HPMRs was to complement the existing network of protected areas, resulting in a network of MPAs with varying levels of protection.

In 2014, the first MCZ in Welsh waters was established around the island of Skomer and the Marloes Peninsula in Pembrokeshire (NRW, 2015). Before 2014 the area had been Wales' only Marine Nature Reserve (MNR) for 24 years. However, Skomer MCZ retained a similar level of protection as when it was a MNR and the HPMR status was not enforced. At the time of writing no area of the Welsh coast was highly protected.

## Study design

The first step in any DCE is to define the good to be valued in terms of its attributes and levels. This study focused on those aspects of MPA network design that were most likely to have an impact on society. Initially, the attributes considered for the DCE were the location, total size of the network, level of protection, proportion of areas with different levels of protection, and the price associated to the enforcement of protective measures. A focus group was carried out with 15 randomly sampled members of the general public to define the final list of attributes to be included in the survey. During the meeting the list of attributes, possible associated levels and alternative formats of the DCE survey

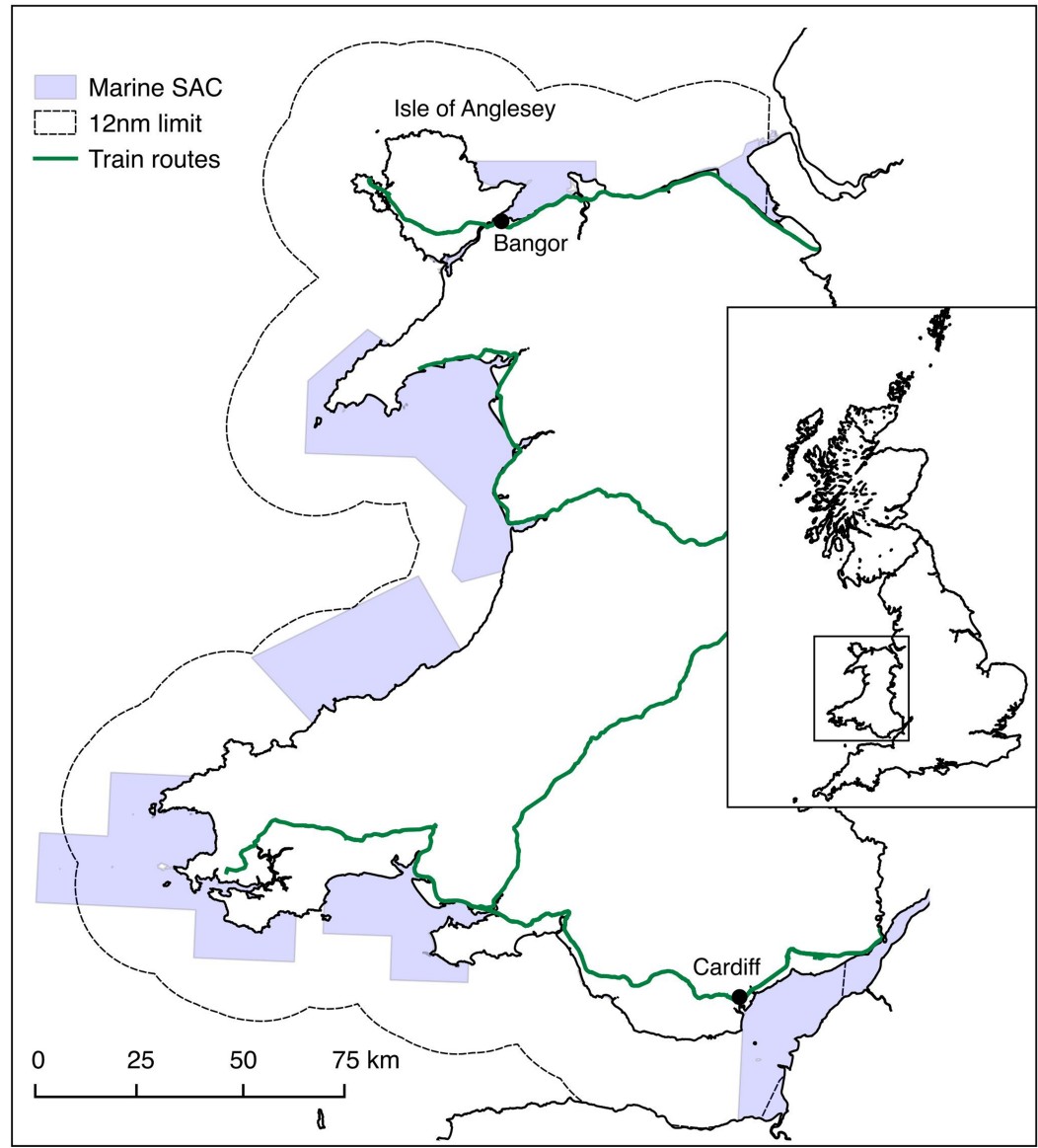

**Figure 1 Overview map of the study area.** Dashed lines indicate the 12 nm territorial waters limit, marine special conservation areas (SACs) are shown in blue, green lines indicate the train routes where questionnaires were undertaken.

were discussed. The focus-group exercise revealed that the full set of attributes was too complex to enable respondents to make meaningful trade-offs during the DCEs. The final set of attributes was reduced to include only size, level of protection, and cost.

The first attribute included in the DCE was the size of the network of MPAs. To define the levels for this attribute, the situation in Wales in 2009 was taken as the baseline. In 2009, 32% of territorial waters were protected under different EU designations with different levels of protection. According to the statutory Governmental conservation advisory body (Natural Resources Wales), it was unlikely that the area of the new network of MPAs would exceed that of the existing protected areas. Thus, the highest level for the

**Table 1 Attributes and levels used in the choice experiment.**

| Attribute | Definition | Levels |
|---|---|---|
| Network size | Percentage of territorial waters to be protected | 10%, 20%, 30% |
| Uses permitted | Uses permitted within the boundaries of the network | – All activities prohibited<br>– Only scientific research and educational activities<br>– Non-extractive activities (i.e., sailing, diving, kayaking, wildlife watching) allowed<br>– Recreational and commercial fishing using non-damaging equipment to the sea floor allowed (previous level included) |
| Cost | Household annual contribution to a neutral charity. The charity works with the government to negotiate, monitor, and manage the MPAs | Payment levels: £5, £10, £25, £50, £100 |

size attribute was set to 30% of Welsh territorial waters (equivalent to 4,826 km$^2$), 20% and 10% were chosen as the alternate levels.

The second attribute in the DCE was the type of protection for the MPA network. In this study four levels of protection were selected as a representation of the most common management alternatives in MPAs: (1) no take zones in which no activities were allowed, (2) areas in which only scientific research and educational activities were allowed, (3) non-extractive recreational activities allowed (e.g., scuba-diving, sailing, kayaking), and (4) recreational and commercial fishing using non-damaging equipment to the sea floor allowed.

The third attribute included in the DCE was a monetary one, which is required to estimate welfare changes of respondents. The range chosen for the monetary attribute and the payment vehicle were determined during the focus group. The final set of selected attributes, their levels and definition are reported in Table 1.

The final questionnaire contained information on the relevance of the marine environment to society from an economic, cultural, and ecological perspective, general information on MPAs, their associated possible outcomes and costs and design issues, and information on the current situation and future plans for Wales. The DCE tasks were located after the general information sections in the questionnaire. In addition to the DCE tasks, information was collected on societal views and attitudes towards MPAs and the environment. Demographic data were collected in order to assess the representativeness of the sample. Average questionnaire completion time was 15 min. A copy of the questionnaire is available through Supplementary Information 1.

## Experimental design and data collection

SPSS Orthoplan was used to generate a ($3^1 \times 4^1 \times 5^1$) fractional factorial experimental design, which created 25 choice options (SPSS Inc, 2008). A blocking procedure was used to assign the options to five bundles of five choice sets, thus five versions of the choice

**Table 2 Choice card example.**

| | Option A | Option B | Current Situation |
|---|---|---|---|
| Size of the network of MPAs | 20% of coastal waters (equivalent to 4½ times the area of Anglesey) | 30% of coastal waters (equivalent to 6¾ times the area of Anglesey) | 30% of coast as SAC (equivalent to 6¾ times the area of Anglesey) |
| Level of protection | Only scientific research and educational activities allowed | Non-extractive activities (i.e., sailing, diving, kayaking, wildlife watching) allowed | Minimum level of protectionMost activities including commercial fishing allowed |
| Cost to you each year | £25 | £5 | No additional cost to you |
| Which of the three options do you most prefer? | I prefer Option A ☐ | I prefer Option B ☐ | I prefer the Current Situation ☐ |

**Table 3 Comparison of respondents' socio-demographic characteristics vs. 2011 census data for Wales (ONS 2012).**

| | Sample average | Census average |
|---|---|---|
| Gender (% male) | 49 | 52 |
| Median age range | 45–59 | 45–59 |
| University degree & above (%) | 63 | 24 |
| Household size | 2.6 | 2.4 |
| Number of children | 0.5 | 1.7 |
| Annual income × capita (£) | 15,248 | 14,129 |

experiments were produced. Each version contained a different combination of five DCE tasks and each choice task consisted of three alternatives (A, B and Current situation in Wales, Table 2).

Data were collected between May–July 2008 using self-completion questionnaires. Questionnaires were administered to consenting passengers on several train routes covering the entire area of Wales. Bangor University research ethics procedures were followed and informed verbal consent obtained from all participants. Since the completion time of the questionnaire was high and required the full attention of the respondent it was felt that trains would offer a receptive audience willing to participate in the study. In the UK, trains are widely used by a cross section of society including business people, students, retired people, and families. Any potential bias that occurred as a consequence of sampling on trains could be assessed through the socio-demographic data collected in the questionnaires (Table 3). Although the chosen survey methodology allowed reaching a broad survey sample, it might have under or over-sampled certain sectors of the population. The problem of sampling hard to reach groups, however, is present in most surveys modes, such as Internet, mail, or telephone interviews.

Two consecutive pilot phases were conducted on a total of 73 respondents prior to the final administration of the survey. Minor corrections to the questionnaires were implemented after the pilots. As the structure of the DCEs tasks did not change during the pilot phases, all pilot questionnaires were included in the final DCE analysis.

**Table 4 Environmental statements included in the survey measured on a five-point Likert scale, ranging from "Completely true" to "Not at all true."**

| Environmental statements |
|---|
| MPAs provide a good way to get the right balance between conservation and activities such as fishing or shipping |
| There are conservation benefits related to MPAs |
| There is no need for MPAs in Wales because the seas around the Welsh coasts are in good health |
| People who are affected by the creation of MPAs, like fishermen, should receive compensation for any financial losses derived from the establishment of MPAs |
| I'm willing to pay higher prices for sea-related products or services to preserve areas of the sea around Wales |
| Costs of MPAs will most likely be greater than the benefits obtained from them |
| MPAs should be large enough to protect every type of organism living in the sea regardless of costs |
| The sea is a common resource and no one should be restricted from using it |
| There is no need to restrict uses that don't damage the seafloor in MPAs |
| Fishing equipment that sits on the seafloor and does not cause damage should be allowed in MPAs |
| Current levels of protection of the sea are enough |
| I like knowing that certain areas of the sea are being fully protected |

A total of 448 people were approached to take part in the study of which 78 declined to participate. Of the 368 questionnaires handed out, 14 did not fully complete the DCE section, leaving a total sample of 354 respondents. Each version of the DCE tasks was allocated approximately 71 times.

## Model specification

Since the interest of the present study was to test for the existence of discrete classes of preference for MPA management within society, we apply LC and RPL models and compare their support. MNL was estimated as a representation of the average preference of the sample since it assumes that preferences are constant across respondents. Models in the study were estimated using mlogit and gmnl R packages (*Sarrias & Daziano, 2017*; *Croissant, 2018*). CE models were designed under the assumption that the observable utility function would follow a strictly additive form. Models were specified so that the probability of selecting a particular MPA configuration scenario was a function of the attributes of that scenario and a constant, which was specified to equal 1 when either alternative A or B was selected, and 0 when the current situation scenario was selected. Attributes size and cost were treated as continuous variables while effects-coding (*Hensher, Rose & Greene, 2007*) was used for the allowed uses attribute. A protected network covering 30% of territorial waters (i.e., Size 30) and the permission of recreational uses within the protected areas were used as a baseline in the models for comparative purposes as this combination is what most closely reflects the current situation.

Socio-demographic and attitudinal variables were included in the models. An "environmental consciousness" factor was calculated according to the responses given for a set of questions presented in Table 4. Factor values ranged from 1 to 4, 1 indicating higher degree of environmental consciousness.

## RESULTS

### Sample characteristics

A total of 354 respondents completed the questionnaire. To assess the representativeness of the sample, socio-demographic characteristics were compared against Welsh population means. Gender distribution, median age range, household size, and average annual income per capita, reflected the distribution in the population (Table 3). However, the proportion of people holding higher education degrees was more than double in the sample than in the population. Conversely, the number of children per household in the sample was lower than in the population.

### Public attitudes towards marine conservation

Results from the attitudinal study revealed that public knowledge regarding MPAs was low. On a scale of 1–4 (1 = "*Never heard of MPAs*" and 4 = "*I consider I've got a good knowledge of MPAs*") 79% of respondents chose either options 1 or 2.

Despite the lack of knowledge on MPAs, the questionnaire showed that the general public had a positive and supportive attitude towards marine reserves. Over 66% of respondents thought that current levels of protection of the sea were insufficient and the vast majority (90% of the sample) liked knowing that certain areas of the sea were fully protected, and agreed with the principle of protection of the Welsh marine environment even if they might never make use of it. Most respondents (75%) agreed that MPAs can provide a good balance between conservation and human activities and a high proportion (86%) thought that there are conservation benefits related to protected areas.

Half of the study participants (50%) believed that the benefits associated with the establishment of protected areas would most likely be greater than its costs. However, in general, it was considered that those affected by the establishment of MPAs should receive compensation for any financial losses (76%) and that paying higher prices for marine-related products or services was a suitable option in order to facilitate the preservation of areas of the sea around Wales (63%). Public opinion was equally divided regarding the proposition that no-one should be restricted from using the sea. Half of respondents (50%) considered that there was no need to restrict uses that do not cause damage to the seafloor, this percentage however dropped to 38% when the specific use under consideration was fishing.

### Determinants of marine protection contribution and latent class preferences

The majority of respondents were able to make a choice between the three alternatives offered in the DCE and only 2% of the sample did not complete the total number of choice tasks. About 76% of respondents were completely, mostly or somewhat certain of the choices they made. One of the two MPA alternatives was chosen 69% of the times and there is evidence that respondents compared the alternatives, as in 84% of the cases respondents varied their choice across the five choice tasks. Only 3% of the sample consistently chose either alternative A or B. Approximately 13% of respondents who

**Table 5 Model fit criterion measures for latent class models with 2, 3, 4 and 5 classes.**

|  | RPL | LC—N classes | | | |
|---|---|---|---|---|---|
|  |  | 2 | 3 | 4 | 5 |
| Log likelihood | −1,275 | −997 | −958 | −932 | −919 |
| AIC | 2,577 | 2,031 | 1,977 | 1,950 | 1,947 |
| BIC | 2,646 | 2,130 | 2,137 | 2,171 | 2,230 |

selected the current situation constantly across the tasks were identified as protesters based on their selection of the "*I support the conservation of the marine environment but object to having to pay for that*" statement. Respondents identified as protesters were excluded from the models as protest responses are inconsistent with the random utility theory framework. Respondents who did not complete all the relevant information sections for the model were also excluded. Models were performed with the remaining 255 respondents. As each respondent undertook five choice tasks, models were run using a total of 1,275 observations.

## Latent class segmentation

Model fit criterion measures, the Akaike (AIC) and the Bayesian information criteria (BIC) were estimated for the RPL and LC models with 2–5 classes to ascertain their suitability (*Scarpa & Thiene, 2005*). Model fit criterion measures indicated that LC models with 2–5 classes presented a better fit than the RPL model. For LC models with increasing number of classes the log likelihood was improved. AIC decreased with increasing number of classes and BIC was at its minimum for the 2-class model (Table 5). No unequivocal decision could be made on the number of classes. Since the greatest improvement in both log-likelihood and AIC was observed when moving from the 2-class to the 3-class model and to facilitate the interpretation of the results by keeping the number of classes to the minimum, the 3-class model was chosen.

## The multinomial logit model

Results from the MNL model (Table 6), representing the average preference of the sample, reflect a significant decrease in utility in the reduction of the area of the MPAs from 30% to 10% of Welsh territorial waters, indicated by the negative sign of the WTP (−£23). In terms of the uses allowed within the boundaries of the MPA, the coefficients for HPMR and MPAs where only research activities would be allowed were significant and negative, indicating an opposition for MPAs with these characteristics (−£54 for HPMRs and −£14 for MPAs restricted to research). The positive sign of the constant shows a preference for MPAs where recreation is allowed. The coefficient for fishing activities within the boundaries of the MPAs was not significant, denoting an indifference towards the permission of these activities.

## The latent class model

Results for the 3-class LCM are given in Table 6, where the upper part displays the utility coefficients for MPAs attributes and the lower part reports class membership coefficients.

**Table 6 Parameter estimates for three-class latent class model. Size 30 and recreational uses have been used as a baseline in the models.**

| | MNL | | | Latent class | | | | | | | | | |
| | | | | Class 1 | | | Class 2 | | | Class 3 | | |
| | Coef. | (s.e.) | WTP | Coef. | (s.e.) | WTP | Coef. | (s.e.) | WTP | Coef. | (s.e.) | WTP |
|---|---|---|---|---|---|---|---|---|---|---|---|---|
| Utility function parameters | | | | | | | | | | | | |
| Const | 1.70 | (0.12)*** | | 1.36 | (0.37)*** | | 3.51 | (0.66)*** | | 4.61 | (0.68)*** | |
| Size 10 | −0.46 | (0.09)*** | −23 | −1.24 | (0.30)*** | −43 | 0.91 | (0.57) | | −0.39 | (0.15)** | −91 |
| Size 20 | −0.14 | (0.09) | | −1.00 | (0.28)*** | −35 | 1.44 | (0.54)** | 13 | −0.03 | (0.14) | |
| HPMR[a] | −1.08 | (0.12)*** | −54 | −2.71 | (0.49)*** | −94 | −2.80 | (0.64)*** | −25 | −0.64 | (0.19)*** | −149 |
| Res | −0.43 | (0.11)*** | −14 | −1.31 | (0.38)*** | −45 | −1.65 | (0.61)** | −15 | −0.00 | (0.18) | |
| Fish | 0.17 | (0.10) | | 0.66 | (0.27)** | 23 | 1.35 | (0.64)* | 12 | −0.48 | (0.17)** | −113 |
| Cost | −0.02 | (0.00)*** | | −0.03 | (0.00)*** | | −0.11 | (0.03)*** | | −0.01 | (0.0)* | |
| Class membership function | | | | | | | | | | | | |
| HE[b] | | | | | | | 0.19 | (0.07)** | | −0.17 | (0.08)* | |
| Acts[c] | | | | | | | 0.24 | (0.19) | | −0.19 | (0.20) | |
| EnvF[d] | | | | | | | −1.27 | (0.21)*** | | −4.14 | (0.30)*** | |
| Inc × capita[e] | | | | | | | 0.00 | (0.00) | | 0.00 | (0.00)*** | |
| LC prob | | | | 0.28 | | | 0.34 | | | 0.38 | | |
| Loglike | −1,382 | | | −958 | | | | | | | | |
| AIC | 2,779 | | | 1,977 | | | | | | | | |
| BIC | 2,815 | | | 2,137 | | | | | | | | |
| N Resp | 255 | | | 255 | | | | | | | | |
| N Obs | 1,275 | | | 1,275 | | | | | | | | |

**Notes:**
[a] HPMR, highly protected marine reserve.
[b] Higher education.
[c] Water related activities (marine).
[d] Environmental factor.
[e] Income per capita.
*** 0.1% significance level,
** 1% significance level,
* 5% significance level.

Membership coefficients for Class 1 were normalized to zero in order to identify the remaining coefficients and all other coefficients were interpreted relative to this normalized class.

The relative size of each class was estimated and each respondent assigned a probability for belonging to each of the three classes. Class membership was determined by the highest probability score. Approximately, 28% of respondents were identified as members of Class 1, 34% as members of Class 2 and 38% as members of Class 3.

Coefficients for the different classes suggest that preferences among classes differed substantially. Costs coefficients were significant for all classes. Members of Class 1 opposed to a reduction in size of the MPA network down to 10% of territorial waters (WTP = −£43) or to 20% (−£35). Class 1 members did not favor HPMRs (−94) or MPAs were only research related activities were allowed (−45). The positive sign of the constant indicates a preference for MPAs where recreational activities were permitted. Similarly, they were willing to pay (£23) in order to allow fishing activities within the boundaries of the MPA

**Table 7 Respondents' profiles for each latent class.**

|  | Class 1 | Class 2 | Class 3 |
|---|---|---|---|
| Within 10 miles % | 49 | 38 | 56 |
| Water activities % | 48 | 47 | 67 |
| High MPA knowledge % | 15 | 21 | 36 |
| Environmental factor | 2.2 | 1.9 | 1.5 |
| Higher education % | 65 | 55 | 80 |
| Income × capita (£) | 16,740 | 16,447 | 18,945 |
| Household size | 2.8 | 2.7 | 2.6 |
| Gender % males | 46 | 42 | 53 |

network. Members of Class 2 were willing to pay (£13) for smaller MPAs which would cover 20% of territorial waters and were indifferent toward a reduction down to 10% of territorial waters. They opposed HPMRs (−£25) and MPAs where only research would be allowed (−£15). Recreational activities and fishing (£12) were supported. Members of Class 3 were indifferent toward a reduction in the MPA network down to 20% but opposed a further reduction to 10% in the network area (−£91). They were not in favor of HPMRs (−£149) but were indifferent toward MPAs where only research would be allowed. They were in favor of MPAs where recreational activities would be allowed but not of MPAs where fishing could take place (−£113). The level of education, income per capita and the level of environmental consciousness showed significant effects on class membership (Table 6). Profiles for the different classes were calculated on the basis of class membership coefficients (Table 7). Members of Class 3 (i.e., against fishing) were characterized by a higher degree of environmental consciousness (i.e., lower environmental factor value, EFV = 1.5), had the highest income per capita (£18,945) and the greatest proportion of members with higher education degrees (80%). Class 3 also showed the greatest proportion of members living within 10 miles of the coast (56%) and undertaking some type of water related activities (67%) in comparison to Classes 1 and 2. The proportion of Class 3 members who considered they had good MPA knowledge was also higher (36%). Members of Class 1 (i.e., bigger MPAs where fishing would be allowed) showed the lowest degree of environmental consciousness (i.e., highest EFV, 2.2), lowest proportion of members with high self-assessed MPA knowledge (15%) and income per capita (£16,740); the proportion of members with higher education degrees (65%) was in between Classes 2 and 3. Class 2 was characterized by the lowest proportion of people living within 10 miles of the coast (38%), of people undertaking water activities (47%), lowest proportion of people with higher education degrees (55%), and lowest income per capita (£16,447). They presented midrange values for MPA knowledge (21%) and environmental consciousness (1.9).

## DISCUSSION

The main focus of this study was to test the existence of preference heterogeneity classes in society for different types and levels of coastal protection in the form of MPAs in a

temperate area. The study provides evidence that the general public supports the establishment of an enhanced network of MPAs in Welsh waters, however, it also shows that societal preferences for coastal MPAs are not homogeneous and that different and defined opinion groups exist. This is in agreement with findings from a similar study carried out in the Northeast United States in which three groups with different preferences for MPAs were identified (*Wallmo & Edwards, 2008*), however, not such evidence exists for European waters. Studies with a focus on Europe have either not assessed preference heterogeneity (*Wattage et al., 2011*) or have done it on an individual basis through the use of Conditional and Random Parameters Logit models (*McVittie & Moran, 2010*; *Börger et al., 2014*; *Jobstvogt et al., 2014*). Studies that have assessed societal heterogeneity on a class level have done it for marine offshore areas (*Kermagoret et al., 2016*; *Börger & Hattam, 2017*) but no studies have so far focused on the coastal zone. In the following discussion, we discuss the validity of the elicited values and the utility of the results for the design of marine management plans.

## Validity of the DCE values

Arguably, the high level of low MPA self-rated knowledge amongst respondents could have hindered the validity of the values elicited from the DCE, since generally DCE encompasses attributes that respondents are familiar with. However, there is evidence that unfamiliarity with particular environmental aspects should not preclude the application of DCE (*Barkmann et al., 2008*) since respondents have been shown capable of learning about unfamiliar aspects during a DCE experiment and to make choices based on their own moral values (*Christie et al., 2006*; *Kenter et al., 2011*). Here, this is supported by the high levels of self-assessed choice certainty and further sustained by the reasonable manner in which certain respondents' characteristics predicted particular choices. As an example, the higher likelihood of a respondent with higher levels of environmental consciousness to prefer MPAs where fishing activities likely to damage the seafloor were banned, shows that a greater concern for the environment is translated into more restrictive management measures. High levels of unfamiliarity with the marine environment have been found in other DCE studies (*Börger et al., 2014*; *Jobstvogt et al., 2014*). However, the focus of these studies was on deep-sea and off-shore areas which, since they are spatially removed from the majority of society, might feel more remote, and unfamiliar than coastal areas. Despite this unfamiliarity, valuation studies are important in highlighting the potential economic values held by the average citizen, which are generally absent from economic assessments (*Hanley et al., 2015*).

The DCE analysis points towards a division of society in classes according to their preferences for MPAs design and management. However, the exploration of socio-demographic data revealed the study sample not to be representative of the Welsh population. Therefore, while the outcome of this study is suitable to be used as a guiding and exploratory tool to achieve MPA designs with higher society acceptance, it should not be used as part of full cost benefit analysis or benefit transfer exercises, as the elicited DCE values are not based on a representative sample.

## Implications for coastal management

Outcomes from studies like the one presented here can be used to shape the development and design of MPA networks on coastal waters and maximize the acceptance and compliance of the associated management restrictions. Results suggest the existence of three distinct classes with different sets of preferences regarding the implementation of MPAs. All classes were in favor of MPAs and were not supportive of the idea of MPAs as HPMRs where no activities could be carried out within their boundaries. Instead, all three classes supported those MPAs where non-damaging recreational activities were allowed. These results align with the concerns expressed by the Welsh public during the public consultation carried out in 2012 on the proposal of highly protected sites around the Welsh coast. Strong opinions were held both for and against the proposed high level of protection. Many were in favor of having such sites but coastal communities and business were concerned about unacceptable socio-economic impacts with little evidence of the benefits (*Welsh Government, 2013*). In addition, it was generally considered unnecessary to have an indiscriminate approach with a high level of protection regardless of whether activities would have an impact on ecological features. In 2014, the Welsh Government established the first MCZ where the HPMR status was finally not adopted.

The main differences between classes arise regarding the size of the network and the permission of fishing activities within their boundaries. We find that two currents of opinion exists, those who are in favor of particular activities within the MPAs (fishing: Class 1 and Class 2) and those who oppose (fishing: Class 3). In terms of the area covered by the MPA network, there was general support for a network that would cover 30% of territorial waters, Class 2 also showed support for 20% of territorial waters while Class 1 was opposed to that idea. In summary, Class 1 favored bigger MPAs where all the activities considered in this study would be allowed and supported fishing to a greater extend than Class 2, Class 2 was in support of both bigger and medium sized MPAs where recreation and fishing would be allowed and Class 3 favored bigger MPAs where recreation but no fishing would be allowed.

In accordance with other DCE studies (*McVittie & Moran, 2010*; *Wattage et al., 2011*; *Börger et al., 2014*; *Jobstvogt, Watson & Kenter, 2014*; *Jobstvogt et al., 2014*; *Kermagoret et al., 2016*; *Börger & Hattam, 2017*) our results indicate that the general public is willing to bear the additional economic cost associated with the implementation of MPAs. However, the comparison of our study, which has solely focused on coastal MPAs, with others which have included inshore and offshore MPA areas, shows differences between society's preferences for management strategies adopted in exclusively coastal MPAs and those that include inshore and offshore areas. As an example, *McVittie & Moran (2010)* showed a WTP ranging from −£17 to £17 for highly restrictive measures in a network of MPAs that include both inshore and offshore areas while results from our study indicate a much stronger opposition to HPMRs located in coastal waters (−£25 to −£149). This highlights the importance of eliciting separate value estimates for coastal and offshore areas, as it would be incorrect to extrapolate values estimated for

offshore areas to coastal zones on benefit transfer exercises. It also serves as an indication that people are capable of making distinctions between the associated society's burden in terms of restrictions between coastal and offshore areas, since the intensity of use of coastal areas by different sectors of society is much greater than for offshore areas, as the latter are generally inaccessible for the majority of people.

Arguably, the design of the DCE in terms of the restriction levels associated with the network of MPAs could have better reflected reality by incorporating zonation within the MPA network. However, the focus groups carried out as part of this study revealed that the cognitive burden imposed by an additional MPA zonation attribute was too great and the design too complex for respondents to make meaningful trade-offs during the DCE. Results indicate that the great majority of the public was not supportive of the idea of MPAs as HPMRs. However, it is possible that the level of support for HPMRs would have increased if the DCE had offered respondents the option of MPAs encompassing areas with a range of different protection and restriction levels. This is supported by the fact that the majority of respondents in the study (90%) indicated that they "like knowing that certain areas of the sea are fully protected" thus, showing their support for areas where no activities are allowed and biodiversity is fully protected. The combination of the LC DCE and the questionnaire results provides useful information for coastal resource managers as it allows to infer that HPMRs combined with adjacent areas with differing levels of user-access, particularly areas where non-damaging recreational activities would be allowed, would appear to be the type of MPA design that would receive the greatest public support, while also ensuring effective conservation. This conclusion is in line with results from a survey carried out among users of MPAs in southern Europe that showed a strong preference for having MPAs with different use zonation, including areas designated for restricted fishing, non-damaging recreational activities, and the full protection of species and ecosystems (*Mangi & Austen, 2008*).

This approach enables decision-makers to evaluate the preferences of those classes with a higher number of members from two complementary angles. On the one hand, the combination of attribute levels that shows the greatest societal support can be identified and pursued, if the MPA design is in line with conservation objectives. In our study, there is indication that MPAs where fishing activities likely to produce disturbance to the seafloor would be banned but recreation would still be allowed, would receive the greatest level of support. On the other hand, it allows for the identification of large groups that might not be willing to engage in imposed restrictions, which would make management and enforcement more difficult (*Ban et al., 2013*). In our study, we have been able to identify that all classes object to the concept of highly restrictive MPAs. Consequently, it would not be advisable for managers to pursue the design of MPAs as exclusively no-take zones, where no type of activity would be allowed. Additionally, through LC analysis it is possible to establish a relation between preferences for particular bundles of attributes and respondents characteristics. Here, we find that those respondents in favor of more restrictive MPAs, where fishing was not allowed, have an overall higher environmental consciousness and posses greater MPA related knowledge. These indications provide coastal resource managers with tools to work towards an increased support for

MPAs where fishing might not be allowed through environmental and awareness education campaigns. The integration of environmental education as part of MPA management (*Zorrilla-Pujana & Rossi, 2014*) is a necessary element in achieving sustainable management, as access to balanced environmental information provides resource users with a wider picture of environmental and societal benefits related to conservation, becoming more willing to accept trade-offs (*Ruiz-Frau, Krause & Marbà, 2018*).

Moreover, these types of approaches provide an opportunity for coastal managers to propose different management measures since society shows an array of divergent interests. Following preference indications, different types of protected areas can be implemented on different coastal areas accompanied by the assessment of societal post-implementation support and compliance to help in the identification of those MPA design combinations potentially most likely to succeed.

## CONCLUSIONS

The attitudes and preferences of resource users of MPAs are a key issue for the management of protected areas (*Jones, 2008*). It has been widely acknowledged that for the management of MPAs to be successful and to ensure compliance it is necessary that users have positive attitudes towards MPAs and their associated regulations (*White, Vogt & Arin, 2000*; *Himes, 2007*). Previous studies have investigated the design of MPAs considering influential stakeholder groups preferences such as fishermen (*Richardson et al., 2006*). Studies which have assessed societal preferences for temperate MPAs have mostly done it for deep-sea and off-shore areas. However, little information has been gathered on societal preferences for MPAs in coastal areas adopting a segmented preference approach. This study shows a general support for the protection of the marine environment in the form of MPAs, however it also shows that there are distinct groups with different preferences for the management of MPAs. We conclude that including this preference heterogeneity in the design of MPA networks in the form of zonation and inclusion of areas which allow recreation but not fishing should be preferred in conjunction with targeted environmental and awareness education campaigns.

### Funding

This work was supported by the Economic and Social Research Council and the Natural Environment Research Council of the United Kingdom as part of a PhD studentship (grant number ES/F009801/01). During part of the write-up of this work Hilmar Hinz was supported by the Ramon y Cajal Fellowship (grant by the Ministerio de Economia y Competitividad de España and the Conselleria d'Educacion, Cultura i Universitats Comunidad Autonoma de las Islas Baleares). The funders had no role in study design, data collection and analysis, decision to publish, or preparation of the manuscript.

## Grant Disclosures

The following grant information was disclosed by the authors:

Economic and Social Research Council and the Natural Environment Research Council of the United Kingdom as part of a PhD studentship: ES/F009801/01.

Ramon y Cajal Fellowship: Ministerio de Economia y Competitividad de España and the Conselleria d'Educacion, Cultura i Universitats Comunidad Autonoma de las Islas Baleares.

## Competing Interests

The authors declare that they have no competing interests.

## Author Contributions

- Ana Ruiz-Frau conceived and designed the experiments, performed the experiments, analyzed the data, contributed reagents/materials/analysis tools, prepared figures and/or tables, authored or reviewed drafts of the paper, approved the final draft.
- James M. Gibbons conceived and designed the experiments, analyzed the data, contributed reagents/materials/analysis tools, prepared figures and/or tables, authored or reviewed drafts of the paper, approved the final draft.
- Hilmar Hinz conceived and designed the experiments, contributed reagents/materials/ analysis tools, prepared figures and/or tables, authored or reviewed drafts of the paper, approved the final draft.
- Gareth Edwards-Jones conceived and designed the experiments, contributed reagents/ materials/analysis tools, authored or reviewed drafts of the paper.
- Michel J. Kaiser conceived and designed the experiments, contributed reagents/materials/ analysis tools, authored or reviewed drafts of the paper, approved the final draft.

## Human Ethics

The following information was supplied relating to ethical approvals (i.e., approving body and any reference numbers):

At the time of the survey (2008) Bangor University (UK) did not require the type of questionnaire used in this study to go through an Ethical Committee Board.

The questionnaire was reviewed by the supervisors of the lead author at the time of her PhD. In the elaboration of the questionnaire and the collection of data Bangor University research ethics procedures were followed and informed consent obtained from all participants.

## Data Availability

Raw data is available in the Supplemental Files. An example of the questionnaire used for data collection is available in File S1. The raw data for the DCE are available in File S2.

## Supplemental Information

Supplemental information for this article can be found online at http://dx.doi.org/10.7717/peerj.6672#supplemental-information.

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
