# Peer review of "Preference classes in society for coastal marine protected areas"

_PeerJ, doi:10.7717/peerj.6672_

## Round 0.1 · original submission · Minor Revisions

Both reviewers have only minor comments and recognize the fact that this is a useful and well prepared contribution. I have made some suggestions for additional literature to consider (but they are just suggestions). One point I feel you really should address is the fact that not everyone is a fan of DCE and at least acknowledge this somewhere.

·

Basic reporting

Introduction

Context to study and background to DCE clearly explained. Objectives clearly stated.

Method

Generally reads well and research approach appears clear though I have to caveat that I’m not an expert in the statistical modelling being undertaken. Questionnaire is supplied in the supplementary information. I noted a few minor statements in the main text that will need checking and might need editing:

Line 130: should this read “random taste variation”? I’m unfamiliar with this terminology.

Lines 190 – 193: check sentence structure. Should line 192 read “level of protection they were offered…”?

Lines 252 – 254: can you show sample locations on Figure 1 Map? Or highlight post code areas on map in terms of where participants came from?

Results

Look comprehensive. Though note that the sample may not be representative of the Welsh population as a whole (the author’s do acknowledge this). I have one specific comment:

Lines 298 – 302: number of children also significantly lower in sample than census data (0.5 vs 1.7), this could potentially have some bearing on the result? Would this be taken into account in the modelling of latent classes?

Discussion and Conclusion

Clearly written and results discussed in the context of the broader literature.

Experimental design

This study used a Discrete Choice Experiment to measure peoples’ willingness to pay for various options of MPA design. The use of a focus group to help define the attributes for the survey will have improved the reliability of the results. The questionnaire appears to be carefully designed, and the authors consider the various factors that may influence the results explicitly in their questionnaire design.

The authors should be commended for administering the number of questionnaires they did. However, the only slight weakness for this study (and this is acknowledged by the authors) is that the sample chosen may not be completely representative or random. This may affect the wider generalisation of the findings to the Welsh population as a whole, though the identification of Latent Classes will still be informative for policy makers making decisions on MPAs.

Validity of the findings

The authors don’t overstate their results and critically appraise the validity of their findings. Sample representativeness, environmental awareness of respondents, and how the DCE questions were asked have all been taken into account in the interpretation of the results. The only omission that came to mind was how much would the outcome of the DCE reflect the time period when the survey was taken?

Additional comments

The authors should be commended on what appears to be a very thorough approach to better understanding people’s willingness to pay for protection of the marine environment, and some of the demographic factors that may influence this. I found the paper interesting to read and the results paint a nuanced and insightful picture of different classes of society's attitudes towards the conservation of the marine environment. The results of this paper will be useful for policy makers making decisions on protection of the marine environment and allow them to rationalise decisions on the basis of the wants of segments of society rather than those of narrow interest groups.

·

Basic reporting

This paper is very well written and covers a very interesting and hot topic - the societal views on MPA designation around Wales. I found it a pleasure to read and review this article. I thought the references used were good however there are other references which could have been included for example:

Ressurreição et al., 2012. (Biological Conservation, 145, pp. 148-159) showed that cultural diversity was just as important as biological diversity when the public are valuing marine biodiversity conservation – the public don’t just value the charismatic organisms the highest. This could be cited around line 77-78.

Sian Rees and colleagues published a nice paper which looked at “The socio-economic effects of a Marine Protected Area on the ecosystem service of leisure and recreation”. Marine Policy, September 2015. This could be cited around line 87.

The paper is very well structured which made it easy for the reader to follow the story that was being told. The figures and tables included were useful and added value to the paper.

Experimental design

The experimental design is sound, with focus-groups and pilots being used to develop and test the attributes before undertaking the actual survey. Using a face-to-face approach on trains is a novel approach but ensures you have a captive audience! I am pleased to see that the limitations of the approach used have been highlighted as papers often neglect these which can lead to the wrong conclusions being drawn by the reader. The sample size is good (n=354 usable responses). I am not a statistical expert but I understood the analysis which had been undertaken and the outputs from it.

Validity of the findings

Despite the data being collected 10 years ago, I think the findings are still valid today and part fill a data gap which currently exists within the UK. The authors recognise the limitations of the survey but nonetheless the findings are valid and could be used to further support the MCZ process in Wales and throughout the UK. The methods employed and the findings could also be applied elsewhere (outside of Wales) and therefore this paper will be a good addition to the literature.

Additional comments

I would have liked to see the findings being discussed in relation to the outcomes of the MCZ process in Wales – this was introduced at the start of the paper however things have progressed since this data was collected. It is actually a shame that this paper was not published earlier as I believe that the findings from this paper align with the concerns that were raised in the public consultation on the proposal to increase the protection of a number of existing MPAs in Wales following the introduction of the MCAA (2009). A few sentences on this in the discussion would add strength to the paper as it would bring things up to date with the MCZ process.

It would also be interesting to see how people’s opinions may have changed over the last decade given for example the impact of Blue Planet II and the plastics issue in raising the awareness of the current threats to our seas. Do you think the general understanding of MPAs will have increased as a result? I appreciate this is outside the scope of your project but it would be very interesting to undertake a similar study 10 years on to see how public opinion has changed.

I have identified a few minor issues that will need to be amended:
Line 47: socio-economic
Line 189: I believe Marine Nature Reserves and SSSIs are UK designations not European designations – please check and amend if required
Line 235: Table 1 (should have a capital)
Line 282: Tables 4 & 5 (should have a capital)
Line 289: ‘Size 30’ – perhaps explain this more clearly in the text as I had to cross reference the tables to understand what you meant by Size 30
Line 391: Sometimes Class is written in italics and other times not – this needs standardising throughout
Line 459: This does not make sense – should ‘and’ be included before ‘the permission’?
Line 681: (2008) SPSS – what does this refer to?

Finally, I would like to add that it is great to see this research published – often data sets that are 10 years old are forgotten about and never written up and published. It is also great to see the research that Gareth was part of still getting published today – great work!

---

## Round 0.2 · accepted · Accept

You have answered all comments made by myself and the reviewers in exemplary fashion.

·

Basic reporting

The author has taken into account the reviewer comments.

Experimental design

The author has taken into account the reviewer comments.

Validity of the findings

The author has taken into account the reviewer comments.

Additional comments

I have suggested to the editor that the article is now accepted based on the revisions made.

·

Basic reporting

Good

Experimental design

Good

Validity of the findings

Good

Additional comments

I am happy that my comments have been taken on board satisfactorily by the authors and fully support publication of this manuscript.